# *vB_CacS-HV1* as a Novel *Pahexavirus* Bacteriophage with Lytic and Anti-Biofilm Potential against *Cutibacterium acnes*

**DOI:** 10.3390/microorganisms12081566

**Published:** 2024-07-31

**Authors:** Xu Li, Wenyan Ding, Zicheng Li, Yi Yan, Yigang Tong, Jialiang Xu, Mengzhe Li

**Affiliations:** 1School of Light Industry Science and Engineering, Beijing Technology and Business University, Beijing 102401, China; lixu@btbu.edu.cn (X.L.); 2230302079@st.btbu.edu.cn (W.D.); lzcyyaxmm@163.com (Z.L.); yanyi@btbu.edu.cn (Y.Y.); 2College of Life Science and Technology, Beijing University of Chemical Technology, Beijing 100029, China; tongyigang@mail.buct.edu.cn

**Keywords:** bacteriophage, *Cutibacterium acnes*, *Pahexavirus*

## Abstract

Acne vulgaris is a prevalent chronic inflammatory skin disease, most common in adolescence and often persisting into adulthood, leading to severe physical and psychological impacts. The primary etiological factor is *Cutibacterium acnes* infection. The overuse of antibiotics for acne treatment over recent decades has led to the emergence of antibiotic-resistant *Cutibacterium acnes* strains. In this study, we isolated and characterized a novel bacteriophage, *vB_CacS-HV1*, from saliva samples. The average nucleotide identity analysis indicated that *vB_CacS-HV1* is a new species within the *Pahexavirus* genus, enhancing our understanding of this underexplored group. *vB_CacS-HV1* demonstrates favorable stability, lacks potentially harmful genetic elements (virulence factors, antibiotic resistance genes, transposons, and integrases), and exhibits potent lytic and anti-biofilm activities against *Cutibacterium acnes* at low concentrations. These advantages highlight *vB_CacS-HV1*’s potential as a promising antibacterial agent that could possibly be complementary to antibiotics or other treatments for acne therapy.

## 1. Introduction

Acne vulgaris (acne) is a highly prevalent chronic inflammatory skin disease worldwide, with peak prevalence during adolescence across all ethnic groups [1,2,3]. It presents as retention acne (open or closed comedones) or inflammatory acne (papules, pustules, and nodules), primarily affecting the face, back, and chest [4]. While acne often starts in adolescence, it can persist into adulthood and progress into severe forms if left untreated [5]. Severe acne can lead to disfigurement, scarring, impaired social functioning, and compromised mental health [2]. Long-term, expensive therapies and dermatological consultations not only impose a significant financial burden on patients but also add to the burdens on the healthcare system and economies [6]. It has been estimated that over 11 million prescriptions are written annually for the treatment of acne [7].

In addition to a genetic predisposition [8,9], environmental factors [10,11], and lifestyle choices [12,13,14,15], *Cutibacterium acnes* (*C. acnes*, formerly known as *Propionibacterium acnes*) is a significant factor closely associated with the onset of acne. This Gram-positive anaerobic bacterium primarily resides in the sebaceous unit, coexisting with *Corynebacterium* species. Emerging research has indicated that the proliferation of *C. acnes* can reduce skin microbiota diversity and enhance keratinocyte differentiation, the disruption of sebaceous gland activity, and local inflammation and immune system activation, which are the four main factors of acne pathogenesis [15,16,17,18,19].

In recent decades, antibiotics have been widely utilized as first-line treatments for acne due to their anti-inflammatory properties. Some commonly used antibiotics include clindamycin, erythromycin, trimethoprim-sulfamethoxazole, macrolides, tetracyclines, amoxicillin, and cephalexin [20]. To enhance therapeutic outcomes, these antibiotics are often employed in combination with non-antibiotic agents. Currently, for acne of all severities, the recommended treatments include 0.1–0.3% adapalene plus 2.5% benzoyl peroxide or 0.025% tretinoin plus 1% clindamycin. In the case of mild to moderate acne, the preferred combination is 3% benzoyl peroxide plus 1% clindamycin [2].

However, the overuse of topical and/or systemic antibiotics, the long treatment courses used for acne, and the availability of over-the-counter (OTC) antibiotic preparations in some countries have led to the worldwide emergence of resistant strains. The study by Coates et al. in the UK, spanning a decade from 1991 to 2000 and involving 4274 acne patients, is the largest investigation on antibiotic-resistant *C. acnes* to date. The study revealed a significant increase in resistance to commonly used anti-acne antibiotics, from 34.5% in 1991 to 64% in 1997 [21]. The review by Clio Dessinioti et al. indicated that multidrug-resistant *C. acnes* is prevalent in East Asia, with resistance rates of approximately 30–50% to clindamycin and erythromycin in South Korea, China, Singapore, and Japan [20]. The study by Zhu et al. in Kunming, China, further reported a high resistance rate of 58% to azithromycin, in addition to clindamycin and erythromycin [22]. These epidemiological data underscore the urgent need to address *C. acnes* drug resistance by enhancing the rational use of antibiotics and developing novel antimicrobial agents.

Bacteriophage (phage) therapy (BT), a promising strategy against antimicrobial-resistant pathogens, is generally considered safe, with a low incidence of adverse events. In recent years, a multitude of studies which have consistently supported the effectiveness of BT in combination with antibiotics against pathogens causing skin and soft tissue infections, such as *Staphylococcus* [23,24] and *Mycobacterium chelonae* [25,26], have been reported. BT targeting *C. acnes* has also shown great potential for future therapies in acne treatment. Recent research by Amit Rimon et al. demonstrated that topical phage therapy in a mouse model with acne-like lesions resulted in significantly improved clinical and histological outcomes [27]. The beneficial effects of using BT to alleviate acne may be attributed to the regulation of keratinocyte apoptosis and the suppression of pro-inflammatory responses [28].

To date, limited research has been conducted using “*Propionibacterium* phage” or “*Cutibacterium* phage” as search terms. Therefore, studies on the isolation and characterization of novel *C. acnes* phages will enhance our understanding of acne treatment and the ecological roles of these phages. ATCC6919 (GenBank accession number CP023676.1), identified as a type IA1 strain of *C. acnes* based on eight-locus expanded multilocus sequence typing (eMLST) [29], is strongly associated with acne and represents a majority of the erythromycin- and clindamycin-resistant strains [29,30]. It is also the most commonly used strain in exploring the pathogenic mechanism of acne [31,32,33], making it an appropriate host for an enrichment-based method of bacteriophage isolation.

In this study, we isolated and characterized a novel bacteriophage, *vB_CacS-HV1*, from healthy saliva samples using ATCC6919, representing a new species within the *Pahexavirus* genus. *vB_CacS-HV1* demonstrated potent lytic activity against *C. acnes*, which is highly associated with acne. Additionally, the phage exhibited effective bactericidal activity against *C. acnes* over a 72 h period at low concentrations, while also efficiently inhibiting *C. acnes* biofilm formation and degrading mature biofilms. These advantages highlight *vB_CacS-HV1*’s potential as a promising antibacterial agent for acne therapy, possibly in combination with antibiotics or other treatments.

## 2. Materials and Methods

### 2.1. Bacteria Strains and Growth Conditions

This study utilized nine *C. acnes* strains, including ATCC6919 and ATCC11827, purchased from Shanghai Microbiological Culture Collection, and seven strains (CAH, CAF2, CAF3, CAH1, CAH2, CAH3, and CAH4) isolated from the facial skin swabs of healthy individuals. All strains were characterized by MALDI-TOF-MS and 16S rRNA sequencing and typed using the PubMLST database. Detailed MLST information is available at the *C. acnes* section of PubMLST (https://pubmlst.org/organisms/cutibacterium-acnes, accessed on 4 July 2024). The strains were cryopreserved at −80 °C in 15% glycerol, unless specified otherwise, and cultured in RCM agar (Haibo Biotechnology Co., Ltd., Qingdao, China) or broth under anaerobic conditions at 37 °C.

### 2.2. Sampling from Healthy Individuals

With verbal consent, saliva samples were collected from the authors and their colleagues. To adhere to ethical guidelines, the participants’ data were anonymized to ensure confidentiality, presenting only the aggregated information. A total of ten volunteers contributed 1 to 2 mL of saliva each, resulting in approximately 15 mL total. An equal volume of SM buffer (50 mM Tris, 10 mM MgSO_4_, and 0.1 M NaCl, with a pH of 7.5) was added to the saliva mixture and thoroughly mixed. Bacteria and cellular debris were effectively removed by centrifugation at 5000× *g* for 15 min, and the mixture was maintained at a temperature of 4 °C. The supernatant was further purified twice by a 0.22 μm membrane (Millipore, Beijing, China) for subsequent phage isolation.

### 2.3. Phage Isolation and Purification

The sample enrichment method was applied to isolate *C. acnes* phages. In brief, 1 mL of the filtrate was combined with 10 mL of the exponential ATCC6919 suspension (10^8^ CFU/mL) and cultured for 24 h at 37 °C under anaerobic conditions. After centrifugation and filtration, the enriched culture was incubated in 10 mL of soft RCM agar (0.6%) with 0.1 mL of the exponential ATCC6919 suspension and then plated. Following 48 h of anaerobic incubation at 37 °C, the plates were inspected for plaques. Clear single plaques were subjected to five successive rounds of purification, yielding the final phage designated as *vB_CacS-HV1*.

### 2.4. Transmission Electron Microscopy

The transmission electron microscopy (TEM) visualization of *vB_CacS-HV1* was performed as in our previous study, though with some modifications [34]. Approximately 20 µL of the phage suspension, with a titer of approximately 10^9^ plaque-forming units (PFU)/mL, was applied to a parafilm. A carbon-coated formvar film on a grid was floated on the phage suspension for 60 s. This was followed by staining with sodium phosphotungstate (pH of 6.8) for 60 s. After the excess was removed, the sample was inactivated by exposure to UV light for 10 min using a 400-mesh grid (Sigma-Aldrich, Beijing, China). Images were recorded by a TECNAI 12 (FEI, Hillsboro, OR, USA) equipped with a 16-megapixel MoradaG3 TEM CCD (EMSIS, Münster, Germany).

### 2.5. Phage Optimal Multiplicity of Infection and One-Step Growth Curve Determination

In order to ascertain the optimal multiplicity of infection (MOI) for *vB_CacS-HV1*, the ATCC 6919 cultures were exposed to the phages at concentrations ranging from 10^−4^ to 10. The phage titer was then quantified by the double-layer agar method, whereby plaques were enumerated under anaerobic conditions at 37 °C for 48 h. This procedure was replicated across all three independent experiments for each MOI. The optimal MOI for *vB_CacS-HV1* was determined by identifying the MOI that yielded the highest titer.

To evaluate the replication characteristics of *vB_CacS-HV1*, a one-step growth curve was conducted. The experimental procedure was performed in accordance with previously established methods. Briefly, the ATCC6919 was cultivated in RCM broth until it reached the exponential growth phase, achieving a concentration of 4.3 × 10^8^ CFU/mL, following inoculation into fresh RCM at a 1% ratio. Subsequently, the *vB_CacS-HV1* (4.3 × 10^7^ PFU/mL) and ATCC6919 (1 × 10^8^ CFU/mL) were combined in 6 mL of RCM broth, maintaining an MOI of 0.1, and the mixture was incubated at 37 °C. Samples of 100 μL were extracted from each mixture at the following specified time intervals: 0, 20, 40, 60, 80, 100, 120, 140, 160, 180, 210, 240, 270, 360, 480, 600, 720, and 840 min. These samples were subsequently diluted and plated using the double-layer agar method to quantify the phage titers. Burst size was determined by dividing the peak phage titer at a plateau phase by the initial infective titer. Each series of trials was meticulously repeated thrice to ensure accuracy and reproducibility.

### 2.6. Determination of Host Ranges and Efficiency of Plating (EOP)

The host range of *vB_CacS-HV1* was evaluated against nine strains representing diverse types and clonal complexes (CC), including six type IA1 (CC1, CC3, and CC4), two type IA2 (CC2), and one type II (CC6) strains. Spot tests were conducted using a double-layer agar method, where the bacterial cultures were mixed with 0.5% RCM agar and overlaid on a 1.5% RCM agar base. The purified phage suspension (approximately 10^8^ PFU/mL) was spotted centrally and incubated at 37 °C for 24 h. Bacterial sensitivity was quantified using a five-point scale based on plaque clarity [35]. The scale ranged from +4 to 0, where +4 indicated clear plaques (complete lysis), +3 represented clearing with a faintly hazy background, +2 denoted substantial turbidity throughout the cleared zone, and +1 signified the presence of some individual plaques. A score of 0 was assigned when no clearing was observed, although a spot might be visible where the pipette tip touched the agar.

The efficiency of plating (EOP) method was employed to further evaluate the phage effectiveness against *C. acnes*, following Kutter’s (2009) protocol [35]. The process involved the serial dilution of phage stock and spotting on different host strain lawns. After incubation, plaque enumeration was performed, and the average PFU count was calculated for each isolate. The relative EOP was determined by comparing the average PFU count on each host to the maximum observed. Host range and efficiency of plating assays were performed in triplicate to ensure result reproducibility and reliability.

### 2.7. Thermal and pH Stability Test

The stability of *vB_CacS-HV1* was evaluated under varying thermal and pH conditions. In terms of pH stability, a phage lysate was combined with 1 mL of RCM broth adjusted to pH levels of between 2.0 and 13.0. The mixtures were then incubated at 37 °C for 1 h, followed by assessing the phage titer using the double-layer agar method.

For the assessment of thermal stability, phage suspensions were subjected to different temperatures (40 °C, 50 °C, 60 °C, 70 °C, 80 °C, and 90 °C) for 1 h each. Post-incubation, the phage titer was determined through dilution and the double-layer agar method.

### 2.8. Genome Sequencing and In Silico Analysis

A 50 mL volume (approximately 10^9^ CFU/mL) of phage suspension was utilized for DNA extraction and sequencing, as previously described [34]. Quality control before and after read trimming was conducted using FastQC (version 0.11.5). Subsequently, the reads were assembled using SPAdes (v3.11.1) following the quality filtering. Annotation was performed via the RAST online service, with gene prediction enabled. The function prediction from RAST was manually verified and corrected using BLASTP (https://blast.ncbi.nlm.nih.gov/Blast.cgi, accessed on 28 November 2023) to search for similar known function proteins of *C. acnes* phages. The tRNA genes were identified using tRNAScan-SE2.0 The phages were assessed for antibiotic resistance genes and virulence factors using the Comprehensive Antibiotic Resistance Database (CARD) [36] and the Virulence Factor Database (VFDB) [37], respectively.

Visualization of the phage genomes was generated using the Proksee online service (https://proksee.ca/tools/prokka, accessed on 14 July 2024). The phage genomes of *C. acnes* belong exclusively to the *Pahexavirus* genus, which encompasses 57 species, according to the International Committee on Taxonomy of Viruses (ICTV). For the phylogenetic analysis of *vB_CacS-HV1*, 63 reference sequences representing these 57 species were acquired from the National Center for Biotechnology Information (NCBI) Virus database. The phylogenetic tree was constructed and visualized using the VICTOR Classification and Tree Building Online Resource server [38], available at https://ggdc.dsmz.de/victor.php (accessed on 26 October 2017). Pairwise average nucleotide identities (ANIs) were calculated for the 20 genomes in this study (with a coverage of >96% and an identity of 88–91% by BLASTN) using the OrthoANI toolversion 0.93.1 [39].

### 2.9. Antibacterial Effect of vB_CacS-HV1 In Vitro

*vB_CacS-HV1* was mixed with ATCC6919 in the logarithmic growth phase at three different MOIs (10, 0.1, and 0.001). The control group contained the same of amount of ATCC6919, and it was adjusted to the same volume with a sterile RCM medium. The mixtures were inoculated into 12 mL sterile cell culture tubes (BKMAMLAB, Changde, China) and incubated anaerobically at 37 °C. After 24, 48, and 72 h, 200 μL of culture from each group was transferred to a 96-well plate (Corning, Beijing, China) for absorbance measurements at 600 nm using a microplate reader (Infinite^®^ M1000 PRO, Tecan, Männedorf, Switzerland) to assess bacterial growth. Simultaneously, 200 μL of the sample was centrifuged at 8000 rpm for 6 min, and the pellet was washed twice with sterile phosphate-buffered saline (PBS) to remove any free phages. The pellet was resuspended in 200 μL PBS, followed by serial dilution and plating to evaluate the antibacterial efficacy of *vB_CacS-HV1*.

### 2.10. Anti-Biofilm Effects of vB_CacS-HV1 In Vitro

Biofilm inhibition by phage *vB_CacS-HV1* was evaluated using a modified crystal violet (CV) staining assay, as previously described [40,41]. ATCC6919 was inoculated into BHI broth (Solarbio, Beijing, China) and incubated anaerobically at 37 °C for 24 h until reaching a concentration of 10^8^ CFU/mL. The bacteria were then mixed with phage *vB_CacS-HV1* at varying MOIs (10, 1, 0.1, 0.01, and 0.001) to ensure a total volume of 200 μL suspension in a 96-well plate, followed by further incubation under the same conditions for 96 h. The negative controls were prepared in BHI without phage *vB_CacS-HV1*. Post-incubation, the supernatants were discarded, and the plates were dried at 37 °C for 50 min. The wells were washed thrice with PBS to remove any planktonic bacteria, with each wash followed by a 10 min drying period at 37 °C. Methanol fixation was performed for 20 min, followed by air-drying at room temperature before staining with 200 µL of 0.1% (*w*/*v*) crystal violet staining (Solarbio, Beijing, China) for 10 min. Excess stain was removed by three rinses with deionized water, and the plates were air-dried. The dye was extracted using 33% (*v*/*v*) acetic acid, and absorbance was measured at 595 nm using a microplate reader.

The degradation of mature biofilms by phage *vB_CacS-HV1* was also evaluated by a biofilm clearance assay [41,42]. Briefly, ATCC6919 was cultured in BHI broth under anaerobic conditions at 37 °C for 24 h to reach a concentration of 10^8^ CFU/mL. The bacterial suspensions (200 μL) were transferred to 96-well plates and incubated anaerobically at 37 °C for 96 h to form mature biofilms. After biofilm maturation, the supernatants were removed and the plates were dried and washed with PBS to eliminate any residual bacteria. Subsequently, 200 μL of phage *vB_CacS-HV1* solutions at varying concentrations (10^8^, 10^7^, 10^6^, and 10^5^ PFU/mL) were added and incubated under identical conditions for 24 h. BHI broth without phages served as a control. The residual biofilm in each group was quantified using CV staining, enabling the assessment of the phage’s biofilm clearance efficacy.

### 2.11. Statistical Analysis

All experiments, including the phage morphology analysis, optimal MOI determination, one-step growth curve, host range and EOP assessments, stability tests, and in vitro anti-bacterial and anti-biofilm assays, were conducted in triplicate. The data are expressed as means ± standard deviations (SDs).

Statistical analyses were conducted using GraphPad Prism version 8 (8.3.0). One-way ANOVA was employed for multigroup comparisons in the biofilm clearance and inhibition assays. Two-tailed unpaired Student’s *t*-tests were used to compare the individual phage-treated groups against the control group of relevant experiments. The findings were deemed statistically significant when *p* < 0.05. Graphs illustrating the results were also generated using GraphPad Prism.

### 2.12. Figure Assembly

The images were cropped and processed in Adobe Illustrator 2021 to assemble the figures.

### 2.13. Data Availability

The phage sequence generated in this study has been deposited in GenBank with the accession number PQ037195.

## 3. Results

### 3.1. Morphological Characterization of Phage vB_CacS-HV1 and Its Plaque Formation

The plaques produced by *vB_CacS-HV1* exhibited varied morphologies (Figure 1A), with three distinct size categories—small, medium, and large. They were classified based on their diameter measurements. Each type was measured by six plaques. The small plaques ranged from 1.0 to 1.6 cm (mean, 1.3 cm ± 0.22), the medium plaques ranged from 2.5 to 3.1 cm (mean, 2.8 cm ± 0.23), and the large plaques ranged from 3.9 to 5.8 cm (mean, 4.72 cm ± 0.66).

The transmission electron microscopy (Figure 1B) of *vB_CacS-HV1* showed phage particles with a mean capsid diameter of 49.16 ± 3.31 nm and a mean tail length of 116.80 ± 8.07 nm, indicating its morphology match within *Siphoviridae*. It should be noted that the term “*Siphoviridae*” is employed in this study primarily to describe the phage’s morphological characteristics. This usage does not imply a taxonomic classification, as the term’s formal taxonomic status has evolved following the 2022 ratification vote [43].

### 3.2. Biological Features of Phage vB_CacS-HV1

*vB_CacS-HV1* demonstrated high efficacy against the host at all tested MOIs, with an optimal multiplicity of infection (MOI) of 0.1, yielding a titer of up to 10^9^ PFU/mL (Figure 2A). The one-step growth curve revealed a latent period of approximately 1 h and an average burst size of 43.8 ± 13.75 PFU/cell under optimal MOI conditions (Figure 2B). The stability tests showed that *vB_CacS-HV1* retained its lytic activity between a pH of 4 and a pH of 11, but lytic activity decreased outside this range (Figure 2C). Additionally, the phage remained stable from 4 °C to 50 °C, with a gradual activity decline between 60 °C and 70 °C and complete inactivation at 80 °C (Figure 2D).

### 3.3. Host Range and EOP of Phage vB_CacS-HV1

Table 1 illustrates the EOP generated by the examined host strains and the lytic activity of phage *vB_CacS-HV1* against these strains quantified using plaque clarity scores. Phage *vB_CacS-HV1* demonstrated lytic activity against all tested strains, as evidenced by the plaque formations (Appendix A). Notably, *vB_CacS-HV1* produced clear plaques against all type IA1 strains, which are strongly associated with acne and comprise most multidrug-resistant isolates.

Within the type IA1 group, which can be further typed into CC1, CC3, CC4, and Singleton (not tested in this study), the phage demonstrated higher lytic efficacy against the CC1 and CC3 strains compared to CC4. Furthermore, *vB_CacS-HV1* displayed variable lytic activity against the type IA2 (CC2) and type II CC6 strains.

### 3.4. Genome Analysis of the Phage vB_CacS-HV1

The genome of phage *vB_CacS-HV1* is a 29,399 bp double-stranded DNA molecule with a GC content of 54.08%. Among the 57 annotated open-reading frames (ORFs), 28 were predicted as functional proteins, while 29 were classified as hypothetical proteins (Appendix A). A majority of these proteins exhibit a high sequence similarity with known *C. acnes* phage proteins, except for ORF22, ORF40, and ORF45, which show no significant sequence similarity (based on the BLASTP database). The *vB_CacS-HV1* genome is devoid of tRNA, virulence factors, antibiotic resistance genes, transposons, and integrases. The ORFs encoding the functional proteins are organized into a highly modular structure along the *vB_CacS-HV1* genome (Figure 3), with modules arranged from the 5′ to the 3′ end, encompassing functions related to DNA packaging, head synthesis, tail synthesis, lysis, DNA synthesis, and regulation.

The packaging module consists of ORF58 and ORF1, annotated as phage small terminase (TerS) and phage large terminase (TerL), respectively, which play pivotal roles in the packaging of viral DNA into the capsid [44].

The phage head assembly module comprises several essential components. ORF2 and ORF4 have been identified as phage portal proteins and head scaffold proteins, respectively, based on their high sequence similarity (>96%) with equivalent genes from the phages *DrParke* and *Ouroboros*. These proteins are crucial for nucleation, a key event in the production of infectious viral progeny [45]. ORF5, ORF6, ORF7, and ORF8 have been annotated as the phage major capsid protein, head-tail adaptor, head closure Hc1, and neck protein, respectively, consistent with their functions in other phage genomes. ORF3 exhibited the highest similarity to two different types of proteins (minor capsid protein and head maturation protease), with higher total scores (>506) and the highest similarity rate (>97.21%). Considering the position of the head maturation protease gene in other *C. acnes* phage genomes and the critical role of the prohead, ORF3 was ultimately annotated as the head maturation protease.

The tail assembly module contains 10 ORFs, including 6 copies of phage minor tail proteins (ORF9, ORF14, ORF15, ORF16, ORF17, and ORF18), 1 copy of a phage tail tube protein (ORF10), 1 copy of a phage tail tape measure protein (ORF13), and 2 copies of tail assembly chaperone proteins (ORF11 and ORF12), which is a typical composition of tail synthesis proteins found in *Siphoviridae* [46].

ORF19 (endolysin) and ORF20 (holin), two components of the lysis module, share a high sequence similarity (>96%) with the query sequence.

The module related to DNA synthesis and regulation was identified within the genomic region spanning from 16,288 bp to 29,062 bp. This region encompasses genes involved in DNA synthesis, such as DNA primase (ORF34), helicase (ORF36), exonuclease (ORF41), endonuclease (ORF55), and dATP/dGTP diphosphohydrolase (ORF42). Additionally, genes associated with regulation, including helix-turn-helix DNA-binding proteins (ORF25 and ORF23), which are well-recognized specific transcription factors, and the phage protein GP33 (ORF35), a transcriptional coactivator for the late genes of the T4 bacteriophage [47], were also identified within this region. Remarkably, a substantial portion of the ORFs in this module—approximately 76%—were annotated as hypothetical proteins, highlighting the potential for further exploration and characterization of their functional roles.

### 3.5. Phylogenetic Analysis of the Phage vB_CacS-HV1

A BLASTn search revealed that the 83 genomes closest to *vB_CacS-HV1*, after excluding partial genomes, are all *C. acnes* phages within *Pahexavirus* (a genus of the class Caudoviricetes), as summarized in Appendix A, with query coverage exceeding 93% and identity achieving over 88%. These findings suggest that *vB_CacS-HV1* belongs to the same taxonomic category.

To determine whether *vB_CacS-HV1* represents a new species, we conducted an ANI analysis on the 16 genomes most similar to *vB_CacS-HV1* (Table 2). The ANI values ranged from 86.65 to 88.92, indicating that *vB_CacS-HV1* is, indeed, a new species within the genus *Pahexavirus*.

The web service VICTOR employs intergenomic distances calculated using the Genome BLAST Distance Phylogeny approach and is well-suited for constructing phylogenetic trees of prokaryotic viruses at the species and genus level using nucleotide sequences. We utilized the D6 VICTOR formula for our phylogenetic analysis. The species-level phylogenetic tree (Figure 4) included 58 well-annotated species within *Pahexavirus*, with reference sequences sourced from the NCBI virus database. Among these species, 52 had single reference sequences each, while 6 species (*Pahexavirus PHL117M00*, *PHL117M01*, *PHL151N00*, *PHL037M02*, *PHL067M01*, and *PHL308M00*) had two reference sequences each, resulting in a total of 63 reference sequences in the phylogenetic tree. The phylogenetic tree results indicated that *vB_CacS-HV1* was considered a new species within *Pahexavirus*, consistent with the ANI analysis result, and that it had a close relationship with *Pahexavirus ouroboros* and *Pahexavirus PHL116M00*.

### 3.6. Antibacterial Effect of vB_CacS-HV1 In Vitro

To evaluate the antibacterial efficacy of *vB_CacS-HV1* against ATCC6919, we monitored bacterial growth using OD600 measurements (Figure 5A) and colony counts (Figure 5B) at the following three time points: 24, 48, and 72 h.

All tested MOI groups demonstrated significant reductions in OD600 values compared to the control groups at the respective time points (*p* < 0.0001). The MOI = 0.001 group showed a continuous decline in OD600 over 72 h, with an approximate 94% reduction at the endpoint. The MOI = 0.1 group exhibited strong antibacterial effects initially at 24 h but displayed fluctuations in the OD600 values between 48 and 72 h, resulting in an approximate 78% reduction at the endpoint. The MOI = 10 group experienced a significant drop in OD600 at 24 h. Although the antibacterial effect persisted at 48 and 72 h, the efficiency diminished over time, culminating in an approximate 20% reduction at the endpoint. In summary, while all MOI groups displayed antibacterial activity, the intensity and consistency varied, with MOI = 0.001 showing the most sustained and effective reduction in bacterial growth.

The results of the bacterial colony counting assays did not entirely align with the OD600 measurements. At 24 h, *vB_CacS-HV1* demonstrated significant antibacterial effects at MOI = 0.001, MOI = 0.1, and MOI = 10. However, at 48 and 72 h, only the MOI = 0.001 group exhibited a significant reduction in bacteria, with a decrease of two orders of magnitude at the endpoint. The MOI = 10 and MOI = 0.1 groups showed no significant difference in antibacterial effects compared to the control groups at both 48 and 72 h.

In conclusion, despite some discrepancies between the bacterial colony counting assays and the OD600 measurements, *vB_CacS-HV1* showed potential as a promising antibacterial agent due to its high bacteriostatic capacity (low bactericidal concentration).

### 3.7. Anti-Biofilm Effects of vB_CacS-HV1 In Vitro

Given that type IA1 strains demonstrate enhanced biofilm formation, a significant factor in *C. acnes* virulence, ATCC6919 (type IA1) was selected as a model to evaluate the antibiofilm activity of phage *vB_CacS-HV1*. This assessment encompassed both biofilm formation inhibition and the disruption of preformed mature biofilms.

In the biofilm formation inhibition assay, the attached biomass was quantified by the ratio of crystal violet staining intensity (OD595) to bacterial growth (OD600), normalizing for growth variations. The inhibition of biofilm formations was observed across all tested multiplicities of infection (MOIs), demonstrating a significant reduction (*p* < 0.0001) compared to the control group after a 48 h incubation period (Figure 6A). No significant differences in the results were observed across the MOIs tested, which ranged from 0.001 to 10.

The mature biofilm degradation was evaluated using a biofilm clearance assay. A 96-h-old biofilm exposed to various phage titers exhibited a significant reduction (*p* < 0.0001) in OD595 compared to the control group after 24 h of treatment (Figure 6B). Notably, the phage titers ranging from 10^5^ to 10^8^ CFU/mL had no significant differences in biofilm disruption.

## 4. Conclusions

In this study, we isolated a novel *C. acne* phage, named *vB_CacS-HV1*, from the saliva of healthy individuals. The *vB_CacS-HV1* bacteriophage exhibits favorable pH and thermal stability, and its genome lacks tRNA, virulence factors, antibiotic resistance genes, transposons, and integrases. The ANI and phylogenetic analyses suggest that *vB_CacS-HV1* is a new species of *Pahexavirus*, thus expanding the understanding of this controversial genus.

The host range determination revealed that *vB_CacS-HV1* can completely lyse type IA1 strains that are highly associated with acne, and it demonstrated a high EOP when these strains were used as host bacteria. The in vitro antibacterial experiments indicated that *vB_CacS-HV1* could eliminate *C. acnes* by two orders of magnitude within 72 h at a low bactericidal concentration (MOI = 0.001). Moreover, *vB_CacS-HV1* was observed to effectively inhibit the formation of *C. acnes* biofilms and to degrade mature biofilms of *C. acnes*. These advantages highlight *vB_CacS-HV1*’s potential as a promising antibacterial agent, possibly complementary to antibiotics or other treatments, for acne therapy.

Based on the currently recognized eMLST typing scheme, *C. acne* can be classified into types IA1, IA2, IB, IC, II, and III [30]. In the host range section of this study, the experiments only assessed the lytic ability of *vB_CacS-HV1* on type IA1, IA2, and II strains, lacking evaluations for types IB, IC, and III. This incomplete analysis of the host range hampers a comprehensive understanding of *vB_CacS-HV1*, highlighting the necessity of obtaining the aforementioned strains in the future to address these gaps. Additionally, *vB_CacS-HV1*’s potential as an antimicrobial agent is currently supported only by in vitro bactericidal experiments. For future clinical applications, further investigations into its safety and efficacy in animal models are crucial.

## Figures and Tables

**Figure 1 microorganisms-12-01566-f001:**
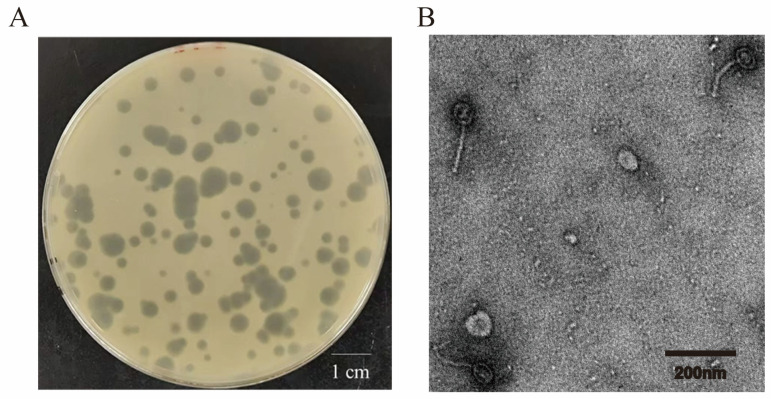
Morphological characterization of phage *vB_CacS-HV1* and its plaque formation. (**A**) Plaques of phage *vB_CacS-HV1* were visualized on a bacterial lawn of the ATCC6919 strain using the double-agar overlay technique. The plates were incubated anaerobically at 37 °C for 24 h. The scale bar represents 1 cm. (**B**) Transmission electron micrograph of *vB_CacS-HV1*. The scale bar indicates 200 nm.

**Figure 2 microorganisms-12-01566-f002:**
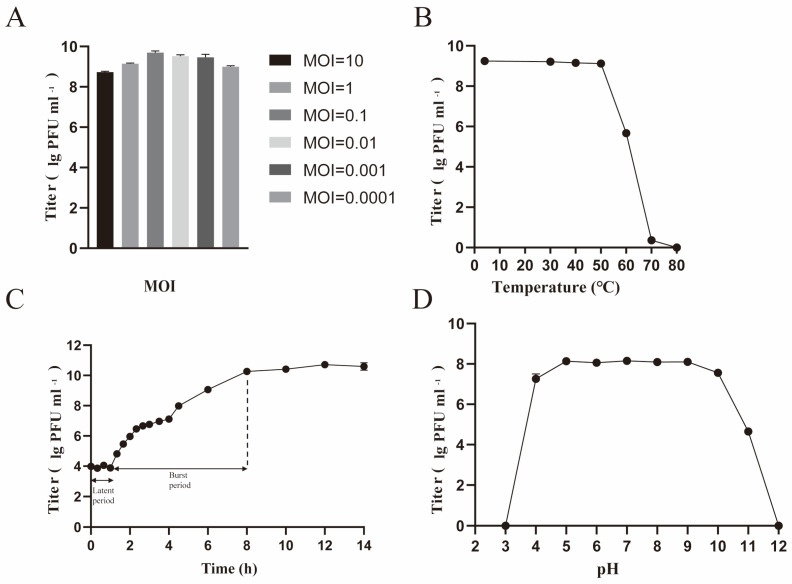
Biological properties of phage *vB_CacS-HV1.* (**A**) The multiplicity of infection of phage *vB_CacS-HV1*. (**B**) Thermotolerance characteristics of the phage *vB_CacS-HV1*. (**C**) One-step growth curve of phage *vB_CacS-HV1*. (**D**) Evaluation of phage *vB_CacS-HV1*’s pH tolerance. The error bars in all figures indicate the standard deviations (SDs) derived from three independent biological replicates (n = 3).

**Figure 3 microorganisms-12-01566-f003:**
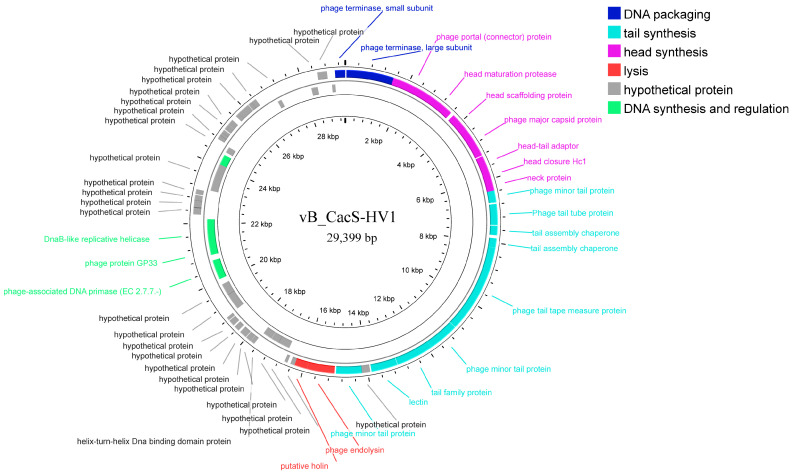
Genome map of *vB_CacS-HV1*. The ORFs are labeled by specific colors according to their annotated function, as follows: red, lysis; dark blue, DNA packing; cyan, tail synthesis; pink, head synthesis; gray, hypothetical protein; and green, DNA synthesis and regulation.

**Figure 4 microorganisms-12-01566-f004:**
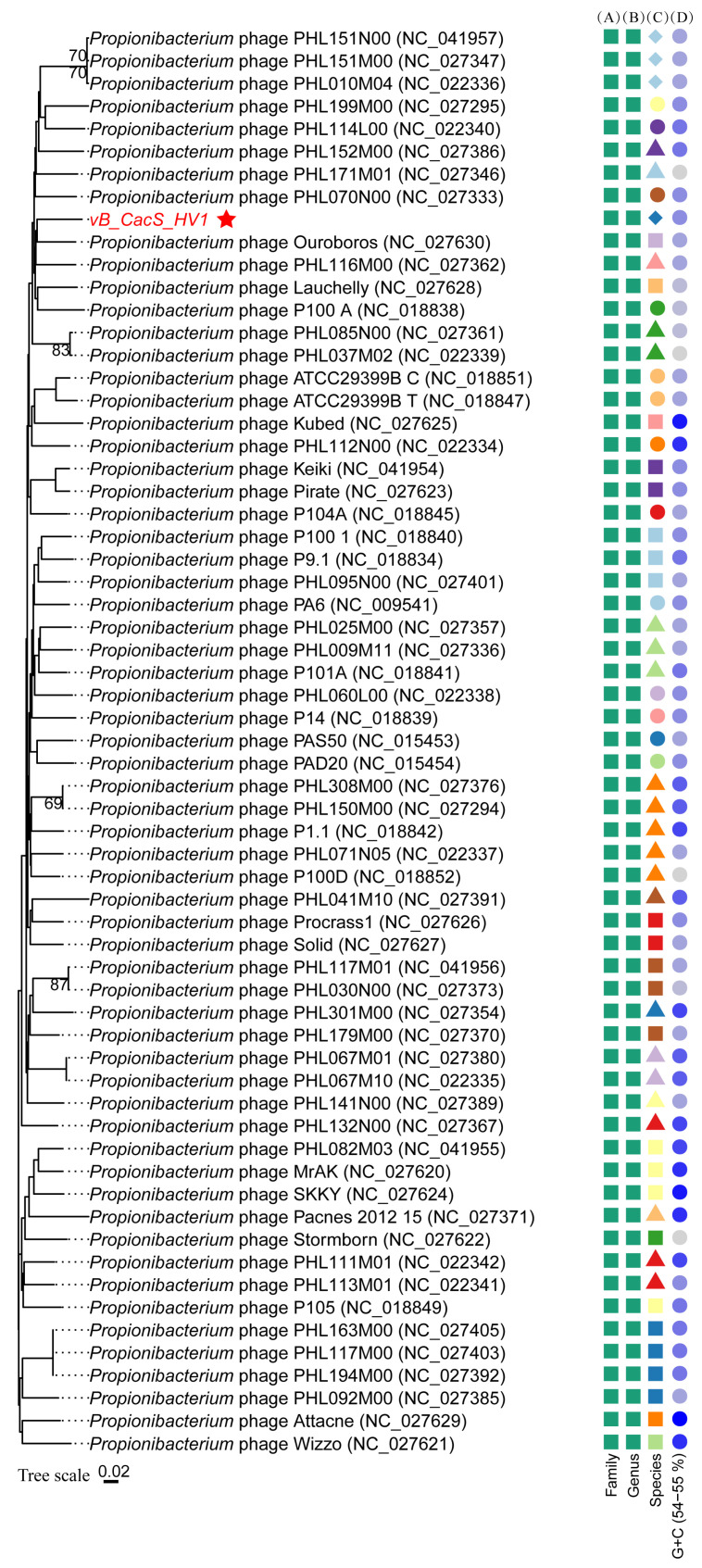
The species-level phylogenetic tree with the D6 formula of VICTOR. The numbers shown above the branches are bootstrap support values from 100 replications, given that branch support exceeded 50%. Annotations are given on the right-hand side of the tree, as follows: family, genera, and species according to VICTOR’s suggested classification by rank (panes (**A**–**C**)), where the combinations of different colors and shapes represent new species. The specific shapes (squares, circles, triangles, and diamonds) do not carry individual meanings. In the genomic G+C content (pane (**D**)), the colors represent the G+C percentage of each sequence, ranging from a minimum of 54% to a maximum of 55%.

**Figure 5 microorganisms-12-01566-f005:**
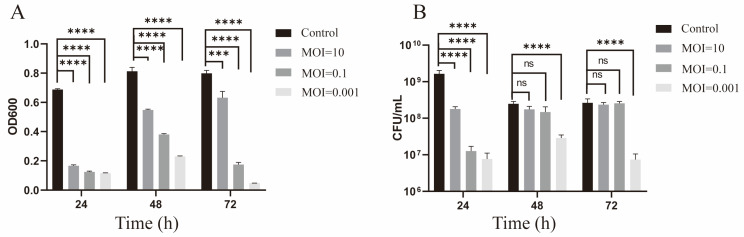
Antibacterial effect of phage *vB_CacS-HV1* in vitro. (**A**) OD600 measurements of the ATCC6919 cultures treated with phage *vB_CacS-HV1* at multiplicities of infection (MOIs) of 10, 0.1, and 0.001 compared to the control group at 24, 48, and 72 h post-inoculation. Error bars, s.d. (n = 3 biological replicates). (**B**) The colony count of ATCC6919 treated with phage *vB_CacS-HV1* at multiplicities of infection (MOIs) of 10, 0.1, and 0.001 compared to the control group at 24, 48, and 72 h post-inoculation. Error bars, s.d. (n = 3 biological replicates). In the above, the asterisks mark statistically significant differences (*** *p* < 0.001 and **** *p* < 0.0001). ns = not significant.

**Figure 6 microorganisms-12-01566-f006:**
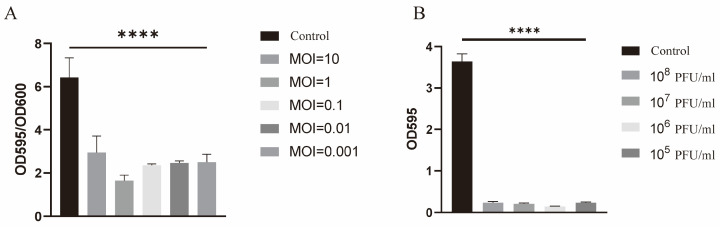
Anti-biofilm effects of phage *vB_CacS-HV1* in vitro. (**A**) The inhibition of biofilm formation by phage *vB_CacS-HV1* at varying multiplicities of infection (MOIs) over 48 h in 96-well plates. (**B**) The degradation of mature *C. acnes* biofilms by phage *vB_CacS-HV1* for different titers over a 24 h period in 96-well plates. The error bars in all figures indicate the standard deviations (SDs) derived from three independent biological replicates (n = 3). The asterisks denote statistical significance (**** *p* < 0.0001). The absence of an asterisk indicates that there was no significant difference.

**Table 1 microorganisms-12-01566-t001:** The host range and EOP of *vB_CacS-HV1*.

Strain	Sequence Type (ST)	Clonal Complex (CC)	Plaque Clarity Score	Efficiency of Plating (EOP)
ATCC6919	1	CC1 (type IA1)	4	0.83 ± 0.14
ATCC11827	1	CC1 (type IA1)	4	0.81 ± 0.17
CAH	115	CC3 (type IA1)	4	0.72 ± 0.25
CAF2	22	CC3 (type IA1)	4	0.69 ± 0.05
CAF3	115	CC3 (type IA1)	4	0.69 ± 0.34
CAH1	115	CC6 (type II)	3	0.10 ± 0.03
CAH2	6	CC4 (type IA1)	3	0.11 ± 0.03
CAH3	4	CC2 (type IA2)	4	0.61 ± 0.10
CAH4	2	CC2 (type IA2)	2	0.12 ± 0.04

**Table 2 microorganisms-12-01566-t002:** Calculations of ANI between *vB_CacS-HV1* and other similar *Pahexavirus* phages.

Scientific Name	Coverage	E Value	Identity	Accession Number	Taxonomy	OrthoANI
*Propionibacterium* phage PAD20	98%	0	88.24%	NC_015454.1	*Pahexavirus PAD20*	86.65
*Propionibacterium* phage PaP11-13	97%	0	89.34%	ON557706.1	*Pahexavirus*	87.91
*Cutibacterium* phage Ristretto	98%	0	89.67%	PP165414.1	*Pahexavirus*	88.75
*Propionibacterium* phage Attacne	97%	0	89.39%	NC_027629.1	*Pahexavirus attacne*	87.53
*Propionibacterium* phage Wizzo	96%	0	88.81%	NC_027621.1	*Pahexavirus wizzo*	87.59
*Propionibacterium* phage Ouroboros	98%	0	91.18%	NC_027630.1	*Pahexavirus ouroboros*	88.47
*Propionibacterium* phage pa27	98%	0	90.85%	MG820634.1	*Pahexavirus*	88.34
*Propionibacterium* phage PHL116M00	99%	0	90.77%	NC_027362.1	*Pahexavirus PHL116M00*	88.62
*Propionibacterium* phage PHL163M00	97%	0	90.75%	NC_027405.1	*Pahexavirus PHL163M00*	88.91
*Propionibacterium* phage PHL117M00	97%	0	90.75%	NC_027403.1	*Pahexavirus PHL117M00*	88.92
*Propionibacterium* phage PHL095N00	98%	0	90.54%	NC_027401.1	*Pahexavirus PHL095N00*	87.98
*Propionibacterium* phage PHL070N00	98%	0	90.38%	NC_027333.1	*Pahexavirus PHL070N00*	87.79
*Propionibacterium* phage PHL092M00	96%	0	90.39%	NC_027385.1	*Pahexavirus PHL092M00*	88.31
*Propionibacterium* phage Procrass1	99%	0	90.36%	NC_027626.1	*Pahexavirus procrass1*	87.99
*Cutibacterium* phage PAP1-1	98%	0	90.31%	OP491959.1	*Pahexavirus*	87.73
*Propionibacterium* phage pa29399-1-D_1	98%	0	90.31%	MG820635.1	*Pahexavirus*	88.12

## Data Availability

The phage sequence generated in this study has been deposited in GenBank with the accession number PQ037195.

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
