# Peer review of "vB_CacS-HV1 as a Novel Pahexavirus Bacteriophage with Lytic and Anti-Biofilm Potential against Cutibacterium acnes"

_microorganisms, 2024, doi:10.3390/microorganisms12081566_

Round 1

Reviewer 1 Report

Comments and Suggestions for Authors

The research conducted by the Authors and the results presented have cognitive value.  Confirmation of these results may also have practical/clinical value. Cutibacterium acnes is the etiologic agent of acne, which is difficult to treat in many patirnts, and long-term antibiotic therapy promotes drug resistance of the bacteria. C. acnes is also responsible for infections in orthopedics. Phage therapy to support the treatment of biofilm-associated infections after joint implantation may contribute to better infection control. 
The paper has the typical layout of an original work. I found no factual or logical errors.

In the Authors' section of the review, I pointed out the lack of information about the limitations of the study, the spelling of microbial names and Latin phrases (e.g., in vitro) - in italics. The correctness of the notation:  ,,et al.'s” I leave to the editor's decision. 

Comments on the Quality of English Language

The authors correctly use scientific terminology. The text is understandable. I did not notice any linguistic errors.

Reviewer 2 Report

Comments and Suggestions for Authors

The manuscript by Xu Li et al., entitled "vB_CacS-HV1, a novel Pahexavirus bacteriophage with lytic and anti-biofilm potential against Cutibacterium acnes" is devoted to the isolation and characterization of a new phage that is active against the causative agent of acne, Cutibacterium acnes. The manuscript is certainly of interest due to the potential application of this bacteriophage in acne therapy. It is well-written and detailed, including a complete genomic characterization.

The manuscript has almost no flaws. However, the main drawback is the absence of sequencing data in GenBank. The authors need to urgently contact NCBI to request an accession number, as the manuscript is already in the review stage, making it difficult to include this number in the finalized text.

Other comments that are easier to address are as follows:

L59: The mentioned review has the reference number 20. Remove numbers 22-25, as they are unnecessary.

L79: 'Extended' in the text, but 'expanded' is used in the reference for MLST. What is the correct method name?

L80: The reference on eMLST is number 33, but it is listed as 34 in the bibliography. Please verify the consistency of references throughout the manuscript text.

L94: Table 1 should appear immediately after its mention in the text, but it is placed several pages later.

L113: 'Brief' should start with a lowercase letter.

L147: 'and' should be placed between 720 and 840.

L197: The '#' at the end of the link can be removed.

L251 and L581: As mentioned above, a valid accession number needs to be inserted. This section can be renamed to 'Data Availability'

L329: Figure S1: There is no explanation of what the figures from A to I mean. What is the difference? Please clarify at least in the legend (L570).

Figure 4: Latin names of bacteria should be italicized. The column (E) can also be removed, as there is no difference visible, just like (A) and (B). Both are just green squares. What is the purpose? For (C) and (D), please explain what the different colors and shapes (squares, circles, triangles, diamonds) represent. It is totally unclear.

Table 2: Again, the Latin names of bacteria should be italicized.

L561: 'Pahexavirus' should be italicized.

L573-577: Authors' names should be shortened to initials.

Comments on the Quality of English Language

English usage is okay.

Round 2

Reviewer 2 Report

Comments and Suggestions for Authors

The authors have addressed all the comments, and the manuscript is now ready for acceptance.

Comments on the Quality of English Language

English quality is okay.